# Instruments to identify risk factors associated with adverse childhood experiences for vulnerable children in primary care in low- and middle-income countries: A systematic review and narrative synthesis

**Winfrida Mwashala**[1]*, **Udoy Saikia**[2], **Diane Chamberlain**[1]

**1** Caring Futures Institute, Flinders University, Adelaide, Australia, **2** College of Humanities, Arts, and social science, Flinders University, Adelaide, Australia

* winfrida.mwashala@flinders.edu.au

**Data Availability Statement:** All data are in the manuscript and/or supporting information files.

## Abstract

Vulnerable children exposed to Adverse Childhood Experiences (ACEs) are lacking visibility in healthcare and social welfare support systems, particularly in countries where there are delays in integrating biopsychosocial care into traditional medical care. This review seeks to identify, evaluate, and summarise existing screening instruments used in measuring risks factors related to Adverse Childhood Experiences (ACEs) in vulnerable children in Primary Health Care (PHC) settings in low- and middle-income countries (LMICs). The target population in this research is children from age (05–18 years) living in poverty and extreme social disadvantage. First, a systematic review was conducted using the Preferred Reporting Items for Systematic Reviews and Meta-Analyses (PRISMA) approach. A mixed-methods narrative synthesis analyzed the studies and instruments used to assess vulnerable children exposed to ACEs. Each instrument was scrutinized for quality, validity, and feasibility for use with vulnerable children in frontline clinical settings. There is a lack of suitable risk assessment instruments to identify biopsychosocial risk factors from exposure to ACEs in vulnerable children in LMIC primary healthcare settings. Among nine identified instruments from the reviewed studies, none were found suitable for rapidly identifying the effects of ACEs. This was due to issues on the reviewed instruments which could hinder their application in the rapid screening of ACEs in frontline clinical settings. This included the, retrospective nature of the instruments, decisional capacity of the rater, institutional capacity in implementation of the instruments and instruments capacity to assess individual risk factors in biopsychosocial dimensions. Therefore, currently, there is lack of instruments that can be used to identify biopsychosocial risk factors of ACEs in vulnerable children in primary care in limited-resource settings. Further development of an instrument for the rapid identification of ACEs in vulnerable children is required for an early recognition and referred for preventive care, treatment, and social support services.

**Funding:** The authors received no specific funding for this work.

**Competing interests:** The authors have declared that no competing interests exist.

## Introduction

Studies suggest that vulnerable children living in Lower and Middle-Income Countries (LIMICs)are at higher risk of exposure to Adverse Childhood Experiences (ACEs) caused by poverty [1–8]. In a study of the Global Early Adolescent Study (GEAS), from 14 countries from LIMIC on, 1,284 adolescents aged (10–14 years) exposed to violence indicated a higher prevalence of ACEs in children living in poor urban neighbourhoods [9,10]. The findings in this study suggested that the early identification and management of ACEs could reduce morbidity and mortality of children at risk [9,11,12]. Despite this understanding, vulnerable children are reported to be poorly identified in Primary Health Care (PHC) settings (see Box 1 for full Glossary of Terms). Little is known about measures available to enhance their early identification for preventive, treatment, and social support services [13–18].

United Nations International Children's Emergency Fund (UNICEF) defines children as individuals below 18 years [19]. During the early years of their life, when children are expected to reach their developmental milestones, they are more susceptible to ACEs exposure due to their young age [8]. The level of vulnerability differs due to cultural diversity and the individual's standard of living and lifestyle. In this review, we are targeting vulnerable children from age (5–18 years). Usually, children are no longer attending under five years of age immunization clinics where their weights and childhood ailments are closely assessed, including malnutrition. Therefore at this stage, they are at higher risks of exposure to ACEs [20]. This review

### Box 1. Glossary of terms

#### Frontline Health Care workers

Initial contact for an ill individual in institutional healthcare facilities. This includes healthcare professionals such as nurses, doctors, and allied health practitioners, through whom patients have initial contact.

#### Primary Health Care (PHC)

Primary health care emphasizes equity of health services accessible to all societies within the communities. It is principled with providing health services focusing on diseases prevention and delivery of health and well-being by focusing on individual needs and preferences. This is conducted by providing a continuum of care from early identification of diseases and hazards to treatment, rehabilitation, and palliative care.

#### Referral pathways

Referral pathways encompass frontline healthcare workers' measures to respond to the newly identified clinical problem. These include preventive, social, and medical support services.

#### Biopsychosocial dimension of risks factors

Variables related to physical and psychosocial factors can predict poor health outcomes for vulnerable children exposed to Adverse Childhood Experiences.

defines vulnerable children as children living in poverty and extreme social disadvantage [21]. This includes children such as those living in poor neighborhoods, humanitarian facilities such as refugee camps, the homeless, and orphans [2,3,7].

Adverse Childhood Experiences have been widely studied for the past two decades [22]. They are defined as repeated traumatic stressors in a child, which affect the normal development of a child's body and brain systems [23]. They are reported to be cumulatively impacting an individual's health, and the higher the exposure of ACEs to a child, the higher the risk of poor health outcomes, including irreversible health conditions [22,24]. Therefore, children who experience poverty-associated adversity are at risk of many long-term developmental, behavioural, emotional, and physical health conditions that affect wellbeing across the lifespan [8,24].

Poverty is defined as deprivation of necessities and is usually measured by the Multidimensional Poverty Indicator (MPI) [25]. When necessities are inadequate, there are higher risks of poor individual health. This includes exposure to social isolation and violence, inadequate access for infrastructures for safety systems, water supply, transportation, and academic prospects [8]. For instance, the impact of the COVID -19 pandemic has increased the risks of undesirable health outcomes in vulnerable children who are at higher risks of exposure to Adverse Childhood Experiences (ACEs) associated with poverty[26–28]. The pandemic has forced 725 million children globally to live in poor socioeconomic households, and 67% of these children are inhabiting developing countries, such as sub-Saharan Africa and South Asia. The rise of global poverty is expected to affect 8% of the total global human population compared to a previous 26% decline of poverty which was made possible through Sustainable Development Goals efforts for the past 20 years [29]. Currently, one billion children are living in multidimensional poverty, and the number is expected to increase by 150 million because of the COVID-19 pandemic [30]. Most of these children are classified as vulnerable. Other factors that exacerbate ACEs in LIMICs result from political instability, ethnic conflict, and natural disasters that lead to social displacement [31].

Inadequate resources for vulnerable children result in food insecurity and malnutrition, drug addiction, neglect, abandonment, and school dropout [32–34]. For instance, the study that assessed the effects of domestic and international remittance of food and food insecurity in LIMIC identified that food insecurity is highly prevalent in Sub-Saharan Africa (55.5%), followed by Latin America and the Caribbean (34.3%) [35]. This is equal to the Multidimensional Poverty Indicator (MPI) report, which estimated that 50 percent of the 1.3 billion people living in poverty live in Sub-Saharan Africa and South Asia [20,30].

There is an interplay between ACEs related to drug addiction and school dropout. Children who are not attending school are more likely to engage in a poor lifestyle and are prone to be exposed to abuse and use illicit drugs [36]. According to UNICEF report on the global initiative on out of school children in–North-Eastern African, countries such as South Sudan, and in East African countries such as Tanzania, Kenya, and Uganda indicate that the problem of drug addiction and school dropout have slightly decreased in the past decade due to various measures aligned with Sustainable Developmental Goals number (03) [36–41]. Yet, this is a problem in some countries from ASIA and other parts of Sub-Saharan Africa due to factors associated with children's involvement in childhood labor which is more likely to expose them to adversities and antisocial behavior. The study conducted in Baghdad with a response rate of 96.2% among 1040 samples confirmed the association between childhood experiences and childhood experiences and substance use in later life [12]. This is quite similar to the study in India that illustrated the prevalence of adverse childhood experiences to be associated with economic challenges, particularly in children living in slums areas and the homeless. This has

forced children of very young ages to be involved with income-generating activities from age 04–5 years [6,42].

Abandoned and neglected children tend to live in poor neighborhoods. Their poor lifestyles and choices exacerbate their susceptibility to communicable diseases such as Tuberculosis and blood-borne diseases such as HIV/AIDS [43,44]. Currently, it is estimated that there are 3.3 million children under 15 years of age living with HIV/AIDS who are more from sub-Saharan Africa [45]. Among these, 35% are not attending health care facilities, have not been diagnosed correctly, and not receiving Anti -Retroviral Treatment (ART) [12,46,47]. Therefore, their exposure to ACEs compromises already weakening life and health discourse [25].

Although vulnerable children require closer medical and social attention, studies indicate that their representation in primary care settings is low [27,28]. These children are invisible mainly due to a high prevalence of social disparities. For instance, financial schemes within the primary care system, such as out-of-pocket funding, are problematic for vulnerable children with inadequate resources [48]. Furthermore, the silos between community stakeholders and primary care result in a lack of formal and reliable existing referral mechanisms [49]. This includes orphanages, Home Based Care services (HBCs) for home visiting services for patients suffering from chronic diseases, including HIV/AIDS, and rehabilitation centers, such as those dealing with substance drug abuse, the sobber houses. The lack of linkage and continuum of care between these service providers is a limitation for healthcare professionals to identify issues that cannot be supported within health care settings. As a result, the lack of care provision diminishes the quality of care and increases stigma for vulnerable children among health care professionals [1].

Studies claim that psychosocial issues are poorly integrated into the current traditional medical model, thus delaying a health risk assessment of diseases related to social factors such as poverty. According to World Health Organisation (WHO), a health risk assessment is a process of assessing and predicting the nature and likelihood of adverse health effects in individuals exposed to the object of the concern. The methodological process of risks assessment begins with the identification of the problem to establish a perspective for the risk assessment and address the consequences that can occur with exposure to adversity, although risk assessment for ACEs is yet in the development process.

Integration of biopsychosocial model of delivery of care into the traditional medical model is delayed in the development process [50,51]. The occurrence is due to structural issues and defragmented health care systems. The biopsychosocial model of care emphasizes the integration of key dimensions of biological, psychological, and social, and environmental effects and how the different aspects of these social and environmental effects interfere with disease and illness. It encompasses more subjective clinical factors of an individual related to the psychological, social, and physical aspects. In contrast, the bio-medical model of care mainly focuses on biological aspects of diseases and illness on curative and treatment measures.

Therefore, it is crucial to identify the instrument that can be used to proactively recognize the risk of ACEs in vulnerable children in primary care in limited-resource settings. The instrument should emphasize early identification of ACEs' risk factors. This ideal instrument will assist in the provision of integrative care and reduce childhood morbidity and mortality by early referrals within primary and acute care and setting [52].

Currently, according to our knowledge, there are no research or healthcare systems that formally screen for early exposures of ACEs in a preventative manner (LIMICs)and refer them before deterioration. Therefore, this research aims to identify screening instruments that identify the risks factors of Adverse Childhood Experiences (ACEs) in vulnerable children in primary care settings. Therefore, the research will answer the following questions:

i.  What are the existing screening instruments for identifying the risks factors of ACEs associated with poverty for vulnerable children in (PHC) settings that include the biopsychosocial dimensions?

ii.  What are their feasibility and applicability?

iii.  Are there sections of instruments that can be adapted to rapidly assess the risk factors associated with ACEs exposure due to poverty?

iv.  What is their validity for use in primary care settings for the identified screening instruments?

## Methods

### Approach

The research team adopted the Preferred Reporting Items for Systematic Reviews and Meta-Analyses (PRISMA) tool and a mixed-methods approach [53,54]. A narrative synthesis approach was chosen to synthesize the selected studies, which had a diverse range of methodological designs, in a structured manner, following the European Social Research Council Guidance on the Conduct of Narrative Synthesis in Systematic Reviews. A review protocol was registered (for blinded review) with the number 192426 within the Prospective Register of Systematic Reviews (PROSPERO) [53,54]. In addition, the research team conducted a narrative synthesis of the current literature that examined the risk assessment instrument available for routine use in frontline clinical settings for vulnerable children exposed to ACEs. A narrative synthesis was useful in the current study since it allows for synthesizing findings from diverse study designs.

The systematic review is conducted in two stages. In stage one, we identified studies that include risk assessment instruments for Adverse Childhood Experiences associated with poverty in vulnerable children. In stage two, we assessed the measurement properties of the risk assessment instrument identified in stage one for their feasibility and applicability for use in primary care settings, using the COSMIN checklist (COnsensus-based Standards for selecting health status Measurement Instruments) [55].

### Eligibility

The searches were performed in two stages. In stage one, we identified the predefined study inclusion criteria [56]. This included i) vulnerable children aged from age 05–18 years to understand their risk factors after early childhood, ii) studies reported on the risk assessment instruments for Adverse Childhood Experiences associated with poverty.

In stage two, we retrieved risk assessment instruments from identified articles in stage one. We selected instruments if they reported on assessment for Adverse Childhood Experiences for vulnerable children aged 05–18. Instruments were selected if they had measurement qualities based on the COSMIN checklist. COSMIN is the measurement property for health-related patient-reported outcomes. That includes reliability, internal consistency, measurement error, content validity, structural validity, interpretability, and responsiveness [55]. Identified studies and instruments were excluded if they were not reported in English and were conducted in the adult population.

### Search strategy

The search strategy was developed with the help of a Higher Degree Research, Librarian at the Flinders University Library, using both MeSH and/or keyword search terms according to the database. A specific search strategy was established based on the PICO outlined in Table 1.

**Table 1. PICO (Population, Phenomenon of interest, Context, Outcome).**

| Population | ("vulnerable child*" or "child* at risk" or "adverse childhood experience") |
|---|---|
| Phenomenon of interest | health indicators AND Social exclusion or neglect or Stigma or Maltreatment or Violence or trauma or Poverty or Kwashiorkor or behavioral risks or drug abuse or addiction or disability or infection or emotional problems |
| Context | developing countries OR Africa OR low-income countries OR Low Middle-income countries, |
| Outcome | (Biopsychosocial effects or outcome Risk Tracking or instrument or assessment or scale or measurements or valid or standard"). |

## Sources of information

The team (WM, US, DC), including Flinders University Librarian, performed a systematic review of the scientific literature. The MEDLINE®, PubMed, CINAHL, PsycINFO, Web of Science, Scopus, and ProQuest databases were searched for articles published from the databases' inception, whereby the earliest recorded were the year 1947 to April 2022. Box 2 outlines the summary of the PRO-Quest search string based on predefined selection criteria.

## Identification of potentially relevant studies

The primary author and the Flinders University research Librarian independently performed a standardized blinded selection process. Articles were managed through EndNote X9 to remove duplicates and COVIDENCE collaboration reference management software to screen for eligibility articles. The titles and abstracts were assessed against the predefined eligibility criteria. The full-text study was retrieved and reviewed for any study whereby the title and abstract were unclear regarding the inclusion and exclusion criteria. After unblinding, disagreements were discussed, and any disagreements were resolved through discussion until a consensus was reached.

## Results of the selection of relevant studies

A PRISMA flow diagram was developed to demonstrate the study selection process [57]. The first stage whereby the articles were reviewed involved the following process,

1. Identification process: The search conducted through both the bibliographic databases and the purposive snowballing technique hand search resulted in 2525 articles. Of these, 1441 were duplicates and were removed, resulting in 1,084 remaining articles.

2. Screening: 1,084 articles were screened for titles and abstracts, with 954 being removed as irrelevant.

## Box 2. Summary of Pro-Quest Search String

("vulnerable child*" OR "child* at risk" OR "adverse childhood experience") AND ((developing countries) OR low-income countries OR Low Middle-income countries) AND (Social exclusion or neglect or Stigma or Maltreatment or Violence or trauma or Poverty or Kwashiorkor or behavioural risks or drug abuse or addiction or disability or infection or emotional problems) AND (Biopsychosocial effects Risk Tracking or instrument or assessment or scale or measurements or valid or standard)

3. Eligibility: Two research team members reviewed 130 articles against the inclusion criteria, with 118 being excluded due to a lack of focus on poverty and adversities being conducted in the adult population or being undertaken in HICs. Fig 1 illustrates PRISMA process of selection of the studies to include in this review.

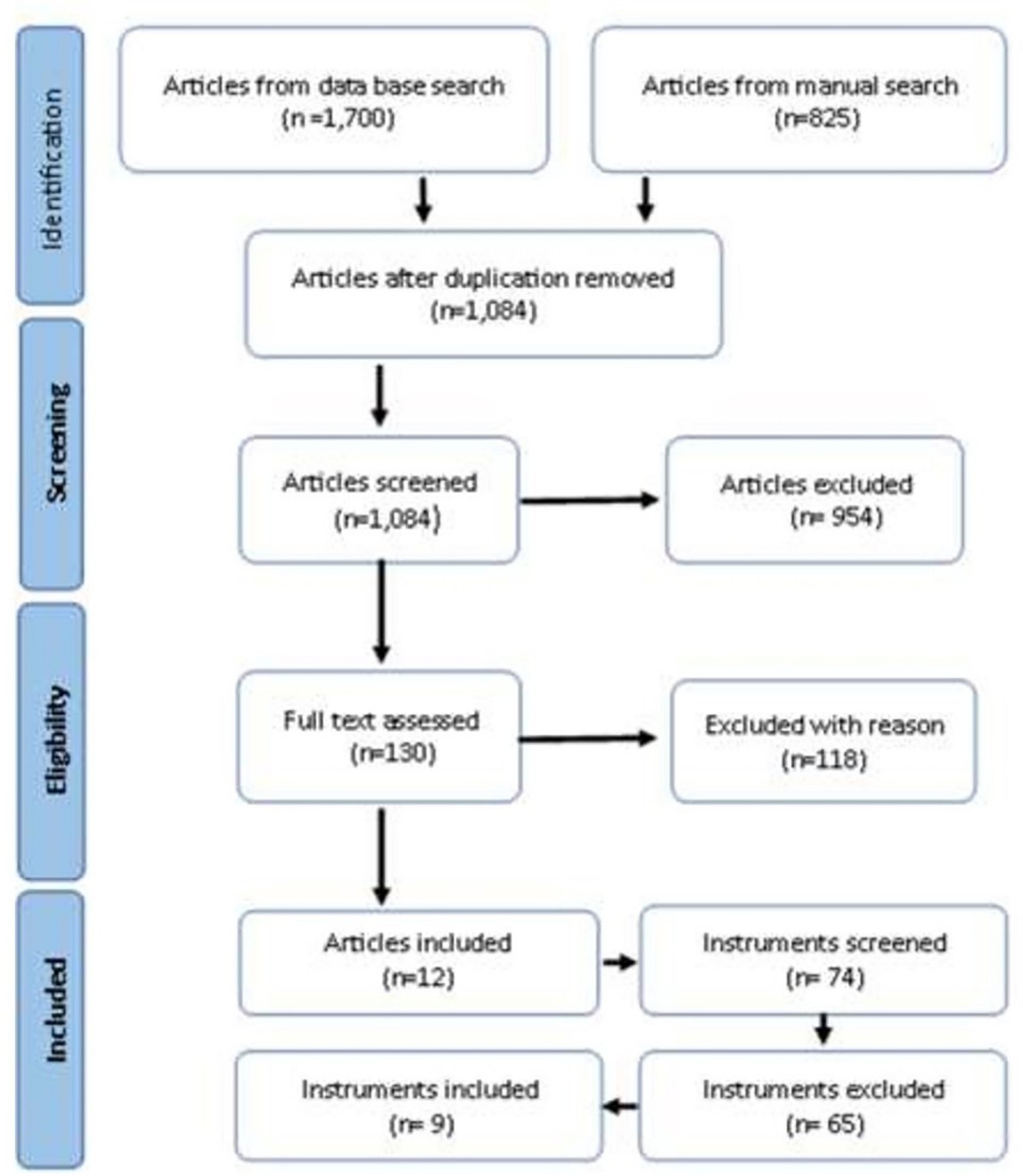

**Fig 1. PRISMA flow diagram.**

**Table 2. A checklist for systematic review critical appraisal—Joanna Briggs Institute.**

| No | Author/ Year | Review questions clearly and explicit stated | Inclusion criteria appropriate for review | Strategy appropriate | Resources and sources used to search for studies appropriate | Criteria for appraising studies appropriate | Methods to minimize errors in data extraction | Methods used to combine studies appropriate | Publication bias assessed | Recommendations for policy or practice | Directives for new research appropriate |
|---|---|---|---|---|---|---|---|---|---|---|---|
| 1 | Nezafat Maldonado, Chandna & Gladstone, 2019 [60] | + | + | + | + | + | + | - | - | + | + |
| 2 | Marlow, Servili & Tomlinson, 2019 [61] | + | + | + | + | + | + | - | + | + | + |
| 3 | Mughai et al.2019 [62] | + | + | + | + | + | + | - | + | + | + |
| 4 | Lee et al., 2018 [63] | + | + | + | + | + | + | - | - | + | + |
| 5 | Semrud-Clikeman, 2017 [64] | + | + | + | + | + | + | - | + | + | + |
| 6 | Saraswathy et al., 2015 [65] | + | + | + | + | + | + | - | + | + | + |

Table 2 the positive (+) sign illustrates, Yes, and the negative (-) sign indicates No.

### Critical appraisal for selected studies /Risk of bias of the selected instrument

The quality of the literature in stage one was assessed irrespectively for risks of bias using the Joanna Briggs Critical appraisal tools and Mixed Methods Appraisal Tool (MMAT), version 2018, for assessing the trustworthiness and relevance of the study designs of the selected articles [58,59]. Three research team members were involved in the assessment (WM, US, DC) and resolved all disagreements. The appraised studies included systematic reviews [60–65], qualitative [66], mixed methods [67–69], and quantitative studies [70,71], as outlined in Tables 2–4. Only two qualitative design studies were excluded, as they could not specify if the interviewed participants were children or adults.

### Data extraction

Two data-extraction forms were created for extracting studies that met inclusion criteria and the instruments. The form for included studies consisted of nine essential items. The descriptive information of the selected studies included:

- First author and the year of publication
- Aim of the study,
- Study area
- Study design

**Table 3. A Checklist for qualitative research critical appraisal—Joanna Briggs Institute analytical tool.**

| No | Author/ Year | Is there congruity between the stated philosophical perspective and the research methodology? | Is there congruity between the research methodology and the research question or objectives? | Is there congruity between the research methodology and the methods used to collect data? | Is there congruity between the research methodology and the representation and analysis of data | Is there congruity between the research methodology and the interpretation of results? | Is there a statement locating the researcher culturally or theoretically? | Is the influence of the researcher on the research, and vice-versa, addressed? | Are participants, and their voices, adequately represented? | Is the research ethical according to current criteria or, for recent studies, and is there evidence of ethical approval by an appropriate body? | Do the conclusions drawn in the research report flow from the analysis, or interpretation of the data? |
|---|---|---|---|---|---|---|---|---|---|---|---|
| 1 | Qi & Wu, 2020 [66] | + | + | + | + | + | + | - | - | + | + |

Table 3 the positive (+) sign illustrates, Yes, and the negative (-) sign indicates No.

**Table 4. A. A checklist for mixed method study using the Mixed Methods Appraisal Tool (MMAT).** B. A checklist for quantitative study using the Mixed Methods Appraisal Tool (MMAT).

| No | Author /Year | Is there an adequate rationale for using a mixed methods design to address the research question? | Are the different components of the study effectively integrated to answer the research question? | Are the outputs of the integration of qualitative and quantitative components adequately interpreted? | Are divergences and inconsistencies between quantitative and qualitative results adequately addressed? | Do the different components of the study adhere to the quality criteria of each tradition of the methods involved? |
|---|---|---|---|---|---|---|
| 1 | Mutenyo et al., 2019 [67] | Yes | Yes | Yes | Yes | Yes |
| 2 | Corcoran & Wakia, 2016 [68] | Yes | Yes | Yes | Yes | Yes |
| 3 | Kidman, Smith, Piccolo & Kohler, 2019 [69] | Yes | Yes | Yes | Yes | Yes |
| | | Are the participants representative of the target population? | Are measurements appropriate regarding both the outcome and intervention (or exposure)? | Are there complete outcome data? | Are the confounders accounted for in the design and analysis? | During the study period, is the intervention administered (or exposure occurred) as intended? |
| 1 | Collings, Vallee & Penning, 2013 [70] | Yes | Yes | Yes | Yes | Yes |
| 2 | Jordans, Ventevogel, Komproe, Tol & de Jong, 2008 [71] | Yes | Yes | Yes | Yes | Yes |

- Sample size
- Age of the participants
- Findings
- Identified health risks
- Summary of the findings

Table 5 outlines a summary of the included study and its characteristics

## Risk of bias of the selected instruments

The identified tools were assessed for the quality of their measurement properties using the COSMIN checklist for use in clinical healthcare settings. These properties were defined based on the COSMIN manual available on the COSMIN website, which included internal consistency, reliability, hypothesis testing, structural validity, measurement error, content validity, cross-cultural validity, criterion validity, and responsiveness. Quality scores or ranks for each property, defined as excellent, good, fair, or poor, were assigned to each property based on a 'worst score counts' basis, as outlined in Table 6 on the risk assessment measurement tools COSMIN checklist analysis. Finally, a narrative synthesis of studies that met the inclusion criteria was conducted. Table 6 COSMIN Checklist summary of the identified risk assessment instrument.

## Data analysis and synthesis of findings

The data were analysed using a mixed-methods approach using recommended guidelines from the Joanna Briggs Institute of Mixed Methods Reviews Methodology Group in 2021. Data synthesis was conducted using a parallel-results convergent narrative synthesis approach of the identified findings following the European Social Research Council Guidance on the Conduct of Narrative Synthesis in Systematic Reviews [56,72]. The evidence was synthesized to establish a meaningful narrative relevant to the research question. Narrative synthesis involves the interpretation of studies from the different methodological approaches which were integrated to interpret findings. Therefore, systematic reviews, qualitative, mixed-methods, and quantitative studies on similar topics that measured childhood adversity in vulnerable children were used to generate current evidence [53]. The analysis involved data searching, listing, describing, and presenting them in tabular form as required. Firstly, the preliminary synthesis was undertaken by describing how the identified instruments were developed and validated for use in assessing childhood adversity associated with poverty in vulnerable and homeless children. They were evaluated using guided questions for their feasibility, practicality, and applicability for routine use in frontline clinical settings. A priori criteria of >80% were used to select potential indicators to develop early warning signs for the rapid assessment of vulnerable children. Subsequently, one author selected proposed indicators to develop a rapid assessment of childhood adversity associated with poverty for vulnerable and homeless children for routine use in clinical settings by identifying the repeated indicators in the study's identified risk assessment tools. Finally, they were recommended for policy, research, and practice on rapid assessment for adverse childhood experiences associated with poverty in limited-resource settings.

## Selection of the instruments

In stage two of the review of the identified instrument, 74 risk assessment measurement instruments were retrieved from the 12 identified studies. The full text of the studies was reviewed to

**Table 5. Summary of the included studies that identified risk assessment instruments.**

| No | First author and the year published | Aim of the study | Study area | Study design | Sample size | Age of the Participants | Findings | Identified Health risks | Summary of the findings |
|----|------|------|------|------|------|------|------|------|------|
| 1 | Nezafat Maldonado, Chandna & Gladstone, 2019 [60] | To develop instruments for risk assessment of mental health | Established to assess children living in LMICs | Systematic review | 300 | 0–19 yrs. | 27 instruments were identified | Externalizing behaviors | Most of the instruments have been designed for high-income countries' health care systems. |
| 2 | Marlow, Servili & Tomlinson, 2019 [61] | To identify Instruments used to screen for autism | Both HICs and LMICs | Systematic review | - | - | Ten instruments were identified | Developmental delays | Instruments need to be refined to identify vulnerable children |
| 3 | Lee et al., 2018 [63] | Review of instruments for measuring exposure to adversity | LMICs | Systematic review | - | 18 years | 32 instruments | Adverse Childhood Experiences | - |
| 4 | Saraswathy et al., 2015 [65] | To assess instruments for early childhood development | LMICs | Systematic review | - | 5 years | - | Disabilities | There is a need for robust, culturally acceptable early child development programs. |
| 5 | Collings, Vallee & Penning, 2013 [70] | To develop and validation of trauma instruments for school-going children | South Africa | Quantitative | 700 | 12–15 | 36 -items | Neglect, violence | This is a retrospective self-administered instrument. |
| 6 | Mutenyo et al., 2020 [67] | To monitor childhood vulnerability | Uganda | Mixed-Method | 832 | - | 2 Instrument | Neglect, HIV/AIDS | Food and nutrition security lacked a robust tracking mechanism |
| 7 | Corcoran & Wakia, 2016 [68] | Using child wellbeing assessments to track progress in family-based reintegration | Kenya and Ethiopia | Mixed-Method | 431 | - | Retrak's model and indicators identified | HIV/AIDS diseases, malnutrition | This tool can guide social workers in the reintegration process. |
| 8 | Semrud-Clikeman, 2017 [64] | To identify measures for the neurodevelopmental assessment of children | LMICs | Systematic review | - | - | Ten instruments | Micronutrient deficiency, Malaria, | Screening measures should not be considered a final comprehensive evaluation of a child's neurodevelopment process. |
| 9 | Jordans, Ventevogel, Komproe, Tol & de Jong, 2008 [71] | The development and validation of the Child Psychosocial Distress Screener instrument | Burundi | Quantitative study | 2,240 | 7–17 | - | Depression | Lay people in the community can use this instrument. |
| 10 | Mughal, et al., 2020 [62] | Review of validated screening instruments for anxiety disorders | LMICs | Systematic review | - | 13–18 yrs. | - | Anxiety | — |
| 11 | Qi & Wu, 2020 [66] | To assess How good are children vulnerability assessment instruments. | China | Qualitative Semi-structured interviews | | - | One tool | - | Instruments need to be well-defined to avoid overlapping and duplication |
| 12 | Kidman, Smith, Piccolo & Kohler, 2019 [69] | Psychometric evaluation of the Adverse Childhood Experience International Questionnaire (ACE-IQ) in Malawian adolescents | Malawi | Mixed-Method | 410 | 10–16 | 1instrument | - | Findings suggest that ACE-IQ is appropriate for use among adolescents from a low-income context |

This table presents a summary of the included studies for identifying risk assessment instruments for Adverse Childhood Experiences associated with poverty in vulnerable children.

**Table 6. COSMIN Checklist summary of the identified risk assessment instrument.**

| Study | Name of the tool | Content validity | Structural validity | Internal consistency | Cross-cultural validity | Reliability | Measurement error | Criterion validity | Responsiveness | Hypothesis testing | Cosmin rank |
|---|---|---|---|---|---|---|---|---|---|---|---|
| **Betancourt et al. 2014 [76]** | African Youth Psychosocial Assessment Instrument (AYPA) | Good | Satisfactory | α = 0.70–0.87 | Good | Satisfactory | N/R | N/R | N/R | Applied | Good |
| **Jordan's et al. 2008 [71]** | Child Psychosocial Distress screener (CPDS) | Good | Good (.81) | α = .5 | Good | Good (.83) | N/R | N/R | Good (.84) | Applied | Good |
| **Van de Hauvel et al. 2017 [94]** | Malawi Developmental Assessment Tool (MDAT) | Good | Good | α = .0.98 | Good | Good | N/R | N/R | N/R | Applied | Good |
| **Su Lyn Corcoran & Joanna Wakia, 2016 [68]** | Child Status Index | Good | Good | N/R | Good | N/R | N/R | N/R | Good | N/R | Fair |
| **Collings et al. 2013 [70]** | Developmental Trauma Inventory (DTI) | | Good | α = 0.70 to 0.81, | Good | Good | N/R | Good | Good | Applied | Good |
| **IPAC (IPEC) - Geneva, ILO, 2014.** | IPAC: The Instrument for Psychosocial Assessment for Child Workers | Good | Good | (and α = .85, α = .91, and α = .81 in three sub-scales). | Good | Good | Good | Good | Good | Applied | Good |
| **Boyes et al. 2019 [92]** | HIV Stigma-by-Association Scale for Adolescents | Good | Good | (α = 0.78–0.87) | Good | Excellent (α = 0.89–0.90 | Good | N/R | Good | Applied | Good |
| **World Health Organisation** | WHOQOL-BREF Quality of life | Good | Good | α = .0.98 | Good | Excellent | Good | Good | Good | Applied | Good |
| **Cluver & Gardner, 2006 [94]** | Strengths and Difficulties Questionnaire (SDQ) | Good | Satisfactory | α = 0.70–0.87 | Good | Good | Good | Good | Good | Applied | Good |

This table presents an assessment of the risk of bias for the identified instruments.

establish meaning and to assess if they met the inclusion criteria. All duplicates and studies with an instrument that measured indicators unrelated to childhood adversity associated with poverty were excluded at this stage, leaving nine risk assessment instruments included for review. Overall, Cronbach's Alpha scores for all scales were above 0.8. This included three risk assessment instruments that assessed the psychological aspects of maltreatment issues on internalizing and externalizing behaviours. Two tools assessed for adversities related to social exclusion and discrimination due to chronic diseases such as HIV/AIDS. Finally, three instruments assessed the physical dimensions of food insecurity and growth and development issues. All studies were randomised to establish statistical reliability and validity.The identified tools were reviewed to evaluate their feasibility, practicality, and applicability for routine use in clinical settings. Three research team members were involved in the assessment (WM, US, DC) and resolved all disagreements. The characteristics of the identified instrument are shown in Table 3.

The descriptive information of the instrument included:

• Author and year

• Study aims

- Instrument/tool name

- Dimensions of scales used

- Indicators

- Psychometric properties

- Target population

- Country and setting

- Rater

- Type of studies

- (SD) for utility measures and 'psychometric properties assessed.'

The descriptive information of the reviewed instrument is illustrated in S1 Table.

## Key findings

This review analysed instruments used to identify the risks of Adverse Childhood Experiences (ACEs) associated with poverty in vulnerable children from age five to eighteen years in primary care settings. The sample sizes of the vulnerable children in the included studies ranged from 300 to 2,240. Geographically, the studies were conducted in LMICs, including China, Malawi, South Africa, Uganda, Ethiopia, Kenya, and Burundi. In all the studies, the ages of the vulnerable children ranged between 0 and 18 years.

Nine risk assessment instruments were included in the review: i) the African Youth Psychosocial Assessment Instrument (AYPA); ii) the Child Psychosocial Distress Screener (CPDS); iii) the Malawi Developmental Assessment Tool (MDAT); iv) the Child Status Index (CSI); v) the IPAC (Instrument for Psychosocial Assessment for Child Workers); vi) the Developmental Trauma Inventory (DTI); vii) the WHOQOL-BREF; viii) the HIV Adolescence Stigma Scale; and ix) the Strengths and Difficulties Questionnaire (SDQ).

These risk assessment instruments measured indicators related to poverty-associated diseases such as HIV/AIDS, malnutrition, and other factors related to violence, natural disasters, and ethnic war. The development of the reviewed risk assessment instruments followed the critical process of i) item generation, ii) psychometric analysis, and iii) theoretical analysis. They were validated for use in LMICs in households and school settings [73,74].

In the initial stage of item generation, the instrument development adhered to either a deductive or inductive methodological process or both. Deductively, tools such as the AYPA, the MDAT, the Stigma Association Index, and the WHO-BREF were created to refine pre-existing scales. Other instruments, such as the CPDS and the IPAC, were created based on qualitative theoretical information from a purposively selected population. Other instruments used both inductive and deductive scale development strategies [74,75].The instrument development and validation processes are explained below.

## African Youth Psychosocial Assessment Instrument (AYPA)

The African Youth Psychosocial Assessment Instrument is a self-report questionnaire for young people that measures psychosocial and behavioural challenges among African children affected by war. This instrument has been developed through a Confirmatory Factor Analysis (CFA) to confirm content dimensions. It has been refined from the Acholi Psychosocial Assessment Instrument (APAI) to the AYPA using item response theory (IRT)to measure internalising and externalising behaviours, such as to conduct and prosocial behaviours (daily

life functioning), depression, and anxiety attitudes. After refinement, the tool consisted of 60 items measured by Likert scale with 1–4 dimensions (rated from 0 = none of the time to 4 = most of the time). Inter-rater reliability and test-retest reliability were conducted during the validity study, identifying good internal consistency of Cronbach alpha α = 0.70–0.87 in each dimension.

The instrument has been used in five studies conducted in Uganda, the Republic of Congo, and Eastern Ukraine to identify adversities related to post-traumatic depression and anxiety related to war-affected syndromes [76–80]. In these studies, children were assessed for post-traumatic stress from violence, family displacement, depression, and anxiety, including antisocial behaviours resulting from political conflict and natural disasters. In these studies, the instrument was also used to assess and analyse how war and climate factors contributed to socio-economic deterioration, contributing to mental health issues and resulting in children dropping out of school [78]. This instrument has mostly been used by researchers and has not been used in healthcare settings by frontline primary health care workers.

## Child Psychosocial Distress Screener (CPDS)

The Child Psychosocial Distress Screener (CPDS) is an instrument that has been developed to curb the mental health effects of associated war issues emerging for children aged 8 to 14 years. It was developed in Burundi, following children being involved in ethnic killings and violence due to political differences and war. It was developed through a culturally grounded methodological approach for use in LMICs settings. The development of the instrument followed four contextual processes to identify the psychometric properties at the community level from high-risk children exposed to distress and their caregivers through semi-structured interviews. The questionnaire measures psychological aspects in four domains: distress, resilience, academic and intellectual capacity, and school attendance.

The instrument provides a baseline reflection of assessment if a child requires further psychosocial management and treatment. The instrument has been used in Ukraine on where children were exposed to traumatic events due to political conflicts and developed Post-traumatic stress disorder (PTSD). In Sri Lanka, the instrument has been used as part of a psycho-educational intervention, while in Haiti, it was used in the context of natural disasters such as earthquakes [80–83]. Furthermore, the instrument has been used in countries that have experienced inter-ethnic conflict, such as Burundi, Sri Lanka, Indonesia, and Sudan in schools to assess maltreatment in war-affected zones in which the rate of dropping out of school increased. The instrument was valid for these settings [84]. This instrument showed a good indication of identifying children for treatment with high-quality consistency and accuracy in test-retest reliability of (.83) kappa coefficient [71].

## Malawi Developmental Assessment Tool (MDAT)

The Malawi Developmental Assessment Tool (MDAT) was developed in Malawi to cater to the need to assess the growth and development of children from birth to six years of age. The instrument was developed by adopting psychometric properties from instruments established in HICs, which were deemed to be 51% culturally inappropriate for the needs of African rural settings [85]. The questionnaire assesses various aspects of early childhood development about fine and gross motor skills, language, and personal-social development to meet the cultural needs of low-resource settings [85]. The instrument consists of 136 items and has been validated in various settings in LMICs for use by healthcare professionals in healthcare settings. In Congo, the MDAT instrument was used to assess 42 children aged from 2 to 24 months (mean = 11.26, SD = 6.37, boys = 22, girls = 20) in four domains of social, fine, and gross

motor skills and language. In this population, the instrument showed good consistency in the social and fine, and gross motor skills variables, while language indicated acceptable consistency (>.5) [86]. The internal consistency and inter-rater reliability of the MDAT were tested using Kendall's Taub test. The data showed positive results with a high number of rτ scores ranging from .923 to .966, illustrating suitable results of inter-rater reliability.

## Child Status Index (CSI)

The Child Status Index (CSI) was developed using MEASURE evaluation with funding from the President's Emergency Plan for AIDS Relief (PEPFAR) to assess children's wellbeing made vulnerable by HIV/AIDS. The instrument has been used in 17 Sub-Saharan African LMICs and Latin America. The instrument was tested for construct validity and inter-rater reliability for the studies conducted in Tanzania and Kenya [87]. The CSI instrument enables consistent case management of the individual child on social support services at the household level. The CSI is a 12 item instrument used to assess the child's needs in six dimensions, (i) food and nutrition (if the child has an adequate supply of food during the year); (ii) shelter and care (if the child is under a guardian or caregiver in the house, and if they have appropriate bedding); (iii) protection (if the child is free from abuse and neglect); and (iv) child health assessed about child access to public healthcare services [87]. Moreover, the instrument also assesses the child's needs, their wellbeing in psychosocial terms, and education to assess the child's established knowledge and skills appropriate to their age. Community volunteers can use it to track the status of children in provisional programs. For instance, in Ethiopia, the CSI instrument was used to measure the effects on children who had been reintegrated into their families from the street using the Retract model to measure the household's needs after family reintegration [68,87]. In a study conducted in Nigeria, 825 orphans and vulnerable children (OVC'S) ranging from 0–17 years, with a mean age of 9.8 ± 4.5 years, were studied to identify their preliminary needs using the CSI index instrument. This study's major indicators identified were food insecurity, maltreatment, and homelessness [88]. In the study conducted in Malawi to validate the CSI index tool as an instrument that could be used to assess vulnerable children find out that the instrument was not valid as there was no consistency between variables, with lower construct validity below (0.7) kappa coefficient [88,89].

## Developmental Trauma Inventory (DTI)

The Developmental Trauma Inventory (DTI) is a tool that consists of 36 items developed specifically to meet the needs of the South African most deprived children population due to socio-economic factors. It is a self-administered instrument created to capture subjective childhood trauma. This includes identifying retrospective childhood trauma related to the post-traumatic syndrome in 10 dimensions related to emotional abuse, community assault, domestic assault, poverty, witnessing community violence, witnessing domestic violence, indecent assault, domestic neglect, rape, and domestic injury. The questionnaire has been developed through open-ended and closed filter questions that probe the child to provide their subjective feelings on their past traumatic experiences. The instrument's internal consistency and construct validity were measured in a study conducted with 802 school students attending a high school located in the Durban metropolitan area during 2011 [70]. In this study, the instrument illustrated higher internal consistency (α = .91), with acceptability in their construct and concurrent validity.

### IPAC (Instrument for Psychosocial Assessment for Child Workers)

The IPAC (Instrument for Psychosocial Assessment for Child Workers) is a manual instrument established to support children at risk due to activities they undertake at a young age to support their families' socio-economic challenges, often undergoing risky work for financial gain. It consists of 48 items intended to identify adversities related to maltreatment due to child labour [90]. The IPAC validity study was taken among children working in the bricks with poverty and psychosocial needs, where laypeople used the questionnaire to assess the needs of the vulnerable children. The instrument was assessed on their cultural dimension for use in different settings. In 38 items which were used in this instrument indicated good reliability for use cross-culturally with only (.10)points of variance in between studied countries (.77) to (.87) internal consistency, with an overall kappa coefficient of (.80), which illustrated that the instrument was good for use [90].

### WHOQOL-BREF

The WHOQOL-BREF is an instrument developed from previous long version instruments that measured quality of life, such as the WHOQOL 100, the SF-12, and the SF-36. During its creation, it was piloted in 15 countries [91]. The instrument has displayed good discriminant validity, content validity, test-retest reliability, and criterion validity in many cultures worldwide. It integrates the good content from the WHOQOL-100 to form 24 dimensions of physical, psychological, socio-psychological, and cultural dimensions that were not available in the previous version [91].

### HIV adolescents' stigma scale

The HIV Adolescents Stigma Scale is a Likert scale instrument, a modified version of the adult stigma scale, validated for teens affected by HIV/AIDS in the United States of America [92]. The previous adult HIV stigma scale had 48 items, while the adapted adolescent scale measures 23 items with good reliability [93]. A modified version of the 23 items has been validated into a seven-item scale to study HIV/AIDS-affected youth in South Africa [92]. In this study, 723 young people participated in the study by completing self-stigma scale measures on two exploratory sub-scales of the experience of being stigmatised and the effects of stigma. The instrument revealed good reliability and validity in the content criterion. As was hypothesised, it was identified that children living in HIV/AIDS-affected communities experienced higher stigma-by-association scores than children in non-affected households [92].

### The Strengths and Difficulties Questionnaire (SDQ)

The Strengths and Difficulties Questionnaire (SDQ) is a three-point Likert scale risk assessment instrument used to measure psychosocial and behavioral factors resulting from poverty-related issues [94]. It has been adapted from HICs to use in LMICs. It is a 25-item scale with symptoms related to emotional issues, conduct problems, lack of attention, hyperactivity, and peer pressure involvement. It has been used in Malawi and translated into the local language, where it was used to assess behavioral problems of children with Severe Acute Malnutrition (SAM), demonstrating good validity (0.73 standardised Cronbach's alpha) [95]. The instrument was also used in a cross-sectional study to assess the development and behavioral challenges resulting from SAM due to the increasing prevalence of HIV/AIDS in Malawian healthcare settings [95].

All instruments were assessed to be applicable in frontline clinical settings. A pre-guided checklist using ten questions guiding a risk assessment tool to use in practice was used. The

developed questions were guided by empirical findings that provided an overview for healthcare professionals and researchers in deciding their instruments of choice for practical use, as discussed in the next section [73].

## The validity, feasibility, and applicability of the identified instruments and recommendations for use in frontline clinical settings

The identified instruments were assessed for validity, feasibility, and applicability for measuring risks factors related to Adverse Childhood Experiences (ACEs) in vulnerable children in Primary Health Care (PHC) settings in low- and middle-income countries (LMICs). Furthermore, they were assessed for the possibility of the sections of instruments that can be adapted to rapidly assess the risk of ACEs for vulnerable children in primary care settings.

Theoretical assessment of the identified instruments followed the process of construct conceptualisation, selection of indicators, and item refinement using Item Response Theory [96]. Most of the instruments used empirical assessment to analyse face and content validity through the use of Confirmatory Factor Analysis to test item construct consistency [74]. The experts engaged in the development of these instruments were mainly from college institutions and mental health departments to ensure that the hypotheses used to establish the instruments were created by experts in the field, and the items constructed would adequately measure the dimensions of the intended hypothesis [75].

The most common psychometric analysis used in assessing instruments were those related to Confirmatory Factor Analysis and Exploratory Factor Analysis. For instance, Exploratory Factor Analysis (EFA) was used in the development of the scale for the HIV Stigma Index, while Confirmatory Factor Analysis (CFA) was used for instruments such as the MDAT, the CPDS, and the DTI. Construct validity and reliability were ensured by using the appropriately required number of targeted children to test the instrument construct using the test-retest reliability technique and inter-observer reliability using Cronbach alpha statistics [74,75].

Their feasibility for use in primary care was assessed by identifying the internal methodological process that underpinned the development of the instrument as well as through external and internal validation measures used. Ten pre-guided questions were used to assess the identified instruments as they have been previously used in criminal justice and forensic psychiatry validation study [73]. The instruments were ranked for their quality, as outlined in Table 6. Ten developed questions were formulated through both internal and external measures to help clinicians evaluate the quality of the instrument, as indicated below.

## External validation

S2 Table outlines the ranking of the identified risk assessment for use in practice using a 10-point checklist for external validation measure.

We used five questions that involved external assessment measures, to assess validation of the identified instrument externally as discussed below.

i. Identify if the tool has been externally validated? This is because tools tend to create good outcome measures if tested for the population they are developed for; hence, it is ideal for validating in a different population. The Child Psychosocial Distress Screener (CPDS) instrument which measured mental health effects related to ethnic war is a good example of an instrument that has been widely tested externally into a different population. It was tested in Burundi, Sri-Lanka, Indonesia, Ukraine, and Sudan for the same population and Haiti for a population with different problems.

ii. Identify if the validation has been conducted in a population of similar interest? The validation of tools needs to be conducted in another population. For instance, the Malawi Developmental Assessment Tool (MDAT) has been validated to conduct risk assessments for children with neuro disabilities and developmental delays. However, this instrument has been used in a separate study to assess the prevalence and severity of developmental and behavioural disorders on a cohort of children admitted to an in-patient nutritional rehabilitation centre. This validation study found that the tool measured accurately, for aims it was created for, with good reliability.

iii. Assess if the tool has been developed based on an efficient methodology? The risk assessment instrument is required to identify the original protocol model used to establish scale and adherence during the validation process. Most of the reviewed instruments did not report the use of the protocol during validation studies. However, they explained the use of the original scale measures and reported modification when required; for instance, in the study which was conducted to refine the dimensional scale measurement properties of the AYPA tool from the AIPI using item response theory (IRT). The instrument was used in a good sample size, which was significant for statistical power. The smallest sample size in the reviewed instrument was 215 children; however, a sound method requires at least 100 representatives for a study to be generalisable. The findings from the validation study need to be published with rigorous methodological details that are replicable.

iv. Assess if the tools report essential information about the conceptual framework? It is a prerequisite that instruments measure standardised measures and discrimination. In the identified tools, the standard measures were analyzed through Confirmatory Factor Analysis (CFA). The logistic regression model was used to predict negative and positive outcome values. The discrimination measures were identified across the instrument using the kappa coefficient to examine consistency.

v. Examine if the tool is feasible and acceptable for use in the scale development objectives? Most of the identified instruments had defined risk dimensions and categories of risk assessment based on the defined symptoms of the child, although they seemed difficult to complete for a child who is vulnerable because of having too many questions. Most of the instruments consisted of subjective indicators using Likert measures, which would be deemed a failure for use with vulnerable children. Most of these children have behavioural issues, including drug abuse. Therefore, self-reported measures can be challenging for vulnerable children rather than interviewing health care professionals. Interviews have been identified as useful in addressing inter-observer bias.

## Internal validation

Risk assessment instruments must have an ongoing internal and external validation process for routine use in clinical practice. There is a need to identify if the instrument has addressed measures as stipulated in S3 Table to be able to recommend its applicability. Questions to be asked to evaluate this include: (i) Is the instrument following a developed protocol? The requirements for an instrument protocol that stipulates study sample, variable measures, expected outcomes, and statistical analysis. Although some of the reviewed instruments did not show the protocol that was used, they provided required information such as (ii) How were the variables selected? (iii) How were the variables weighted? (iv) How were other parameters selected? And (v) Has internal validation been undertaken? S3 Table outlines the ranking of the identified risk assessment for use in practice using a 10-point checklist for internal validation measure.

### Assessment of the identified instruments on applicability and recommendations for use in frontline clinical settings

Six aspects of the selected instruments were assessed to identify their applicability for use. This included an evaluation to identify the dimensions of the indicators used within the instruments, health risks addressed, rater and time used, scalability, and feasibility. S4 Table outlines how the instrument was assessed for its applicability for use in frontline clinical settings. No instrument was identified as feasible for use in frontline clinical settings.

### Biopsychosocial dimension

In this review, the associated factors for adversities and poverty created a range of challenges, such as poor psychological and physiological health outcomes, which needed risk assessment for early identification and treatment. Unfortunately, some of these tools were not assessed in clinical settings. Instead, they were developed by researchers and social workers mostly in home-based care settings, which missed out on homeless and vulnerable children's indicators. These instruments captured reflections on physical factors, such as diseases, psychological factors (emotional, individual behavior, etc.), social factors related to climatic disasters, war, crime, and socioeconomic stress, and how they are associated with vulnerable and homeless children.

### Health-related risks

The reviewed instruments assessed health risks related to adversity due to poverty. These instruments captured reflections on issues related to physical factors, such as diseases, psychological factors (emotional, individual behaviour, etc.), and social factors related to climatic disasters, war, crime, and socioeconomic stress, and how they are associated with vulnerable children.

### Rater/Time

The risk assessment instruments were mainly designed to be used by children to rate themselves or with support from caregivers/parents or teachers. Therefore, there might be a challenge in the use of vulnerable and homeless children. The time taken for the interviews was too long for use in clinical settings. Studies indicate that in LMICs, healthcare professionals usually have busy schedules and an overwhelming number of patients; therefore, there is a need for risk assessment measures that require less time and are easily available.

### Scaleability

The scales had good reliability to track biopsychosocial health risks related to poverty and adversities, and they were consistent in their psychometric properties' constructs. In the studies undertaken for validation, it was identified that most of the tools established good coefficient kappa for construct, content, and criterion validity. However, due to socioeconomic variance, most of these instruments required further validation studies to assess different populations, such as those living in urban and rural settings.

### Feasibility

Screening for ACEs associated with poverty-related conditions in healthcare systems is deemed to reduce government expenditure in addressing chronic conditions, which can happen later in life because of adversities faced during childhood. The identified instruments for this review were mostly accessible online; however, the challenges were for users in rural

settings with less Internet access. Also, most of the instruments required validation before use in different settings, requiring further resources for many studies.

## Cost-effectiveness/availability

Screening for ACEs associated with poverty-related conditions in healthcare systems is deemed to reduce government expenditure in addressing chronic conditions, which can happen later in life because of adversities faced during childhood. The identified instruments for this review were mostly accessible online; however, the challenges were for users in rural settings with less Internet access. Also, most of the tools required validation before use in different settings, requiring further resources for many studies. S4 Table outlines how the instruments were assessed for their feasibility, biopsychosocial dimension, and their applicability for use in frontline clinical settings.

## Indicators for use in clinical settings for rapid identification for vulnerable children exposed to ACEs

Based on the identified instruments, there was a remaining question to ask on what patterns of indicators need to be used for early screening of vulnerable children exposed to ACEs in healthcare that address biopsychosocial dimensions of ACEs associated with poverty?

According to Georg Engel, there is no single cause-effect model to illustrate clinical experiences. In his theory, he argues that biomechanical and social complexities are involved in the causality of diseases [97]. There are three dynamics in disease causality trajectories, i) Complexity and causality lead to no single micro-organism being the cause of disease; rather, it is the interplay of several factors. For example, a vulnerable and homeless child living on the street can lead them to acquire behavioural problems such as drug addiction and exposure to blood-borne diseases, including HIV/AIDS; ii) Structural causality involves multiple pathways relative to the causality of illness. The effects of poor neighbourhoods with a lack of basic needs such as water and infrastructure can be a leading cause of under-nutrition in vulnerable children such as the homeless children, resulting in congestive heart disease due to kwashiorkor; iii) Interpretating effects and causality. Consequently, it would have been ideal for the instruments to have monitored the impact of ACEs by addressing biopsychosocial dimensions and illustrating the multiple pathways of how the illness occurred and the overall management plan [98]. Therefore, based on the biopsychosocial dimensions of how physical, biological, social, and environmental factors are relative to illness's causes and effects, we identified patterns of indicators related to ACES on the identified instruments. S5 Table illustrates risk assessment indicators identified in the reviewed instruments.

The identified instruments had similarities in the risk assessment indicators. The differences were due to the nature of adversity, which the instrument was established to measure. The selected indicators were given a score of 80% after appearing in more than five tools. We selected similar indicators across the identified instruments and proposed them for further use and development of the rapid assessment of the risk factors for childhood adversity associated with poverty in clinical settings, as outlined in S2 Table. This included, i) thinking/learning/concentration, ii) Sadness iii) Withdrawal, iv) Anxiety, vi) Homelessness viii) Income ix) Access to healthcare x) Fatigue. S6 Table shows selected indicators (Proposed for) rapid assessment of the ACEs associated with poverty for vulnerable children.

## Discussion

Nine instruments were identified in this review that measured the effects of ACEs in children affected by adversity related to poverty, climatic challenges resulting in social displacement,

and ethnic conflicts. Although the identified instruments have methodological strength, they are not explicitly designed to use in a frontline clinical setting. None of the identified measuring instruments in its original form could be used to screen in a rapid form the risks of Adverse Childhood Experiences (ACEs) in vulnerable children in primary care settings [11,69]. Furthermore, there were issues identified in the reviewed instruments, which could hinder their application in the rapid screening of ACEs in frontline clinical settings. This included ii) The retrospective nature of the instruments ii) The decisional capacity of the rater iii) The Institutional capacity in implementation of the instruments iv) Poor instruments capacity to assess individual risk factors in biopsychosocial dimensions. The identified instrument was discussed below concerning the linked issues:

## The retrospective nature of the instruments

Capturing the risk factors that could cause these children's illness in rapid form is not accessible using the currently reviewed instruments. The primary care systems were identified to lack a standardized rapid identification assessment system, and measures that could recognise and assess the risk of ACEs in vulnerable children for preventive care and referral to social support and acute care services [48]. Most of the reviewed instruments, including the standard screening instrument for ACEs scores, have focused on identifying retrospective childhood adversity experiences [99]. There is little information to address childhood adversity on recognition measures through rapid, observable assessments [100,101]. The items in the reviewed instruments were not intended to be used as rapid assessment indicators or measures. Therefore, the early risks signs of deterioration for vulnerable children's health conditions are currently missed. In Malawi, a study was conducted to examine the psychometric properties of the World Health Organisation ACE-IQ instrument, among rural adolescents affected by HIV/ AIDS. This study confirmed that the factor structure of World Health Organisation ACE-IQ instrument on its validity and reliability could be adapted for use in LMICs [102]. The problem is the retrospective aspect of this instrument. This instrument excludes the rapid aspect of the children's risk assessment needs, especially for vulnerable children who might need immediate medical and social support services [69]. Furthermore, the World Health Organisation (ACE -IQ) instrument has identified adverse childhood to be associated with key factors such as childhood abuse, neglect, and household dysfunction retrospectively [69]. The childhood adversity risks in LIMIC can result from underlying poverty issues that need preventive biopsychosocial measures. This is because their effects can result in irreversible health outcomes [103,104].

## The decisional capacity of the rater

Engagement of children and their families in the process of assessment to identify risk factors of their diseases has been widely promoted in clinical settings [105]. In the current review, instruments were designed for self-reporting with the help of caregivers. In most cases, vulnerable children such as those who are homeless are usually on their own without a caregiver. As well, their capacity to comprehend the information might be limited due to a lack of literacy and poor comprehension due to behavioural issues such as drug addiction. Therefore, there is a need to revaluate risk assessment measures related to vulnerable children [105].

## Institutional capacity in implementation of the instruments

The instruments identified in this review had multi-items, which were deemed unsuitable for use in the primary care settings by frontline health care professionals due to the scarcity of physicians. According to WHO reports, there is an overwhelming ratio of physicians to

patients of 0.7 doctors per 1000 population in developing countries [106]. Hence, the more items in the risk assessment instrument, the less likely it is to attract users. Furthermore, many instruments did not report on access to the manual guidelines used for scale development, which hinders replicability in other settings. Despite studies suggesting that instruments with many items are reliable due to their higher statistical significance, the nature of the clinical setting in LMICs is that they are busy with an overwhelming number of patients and very few healthcare professionals. For instance, two instruments had more items than others: Malawi Developmental Assessment Tool (MDAT), which had one hundred and thirty-six items, and African Youth Psychosocial Assessment Instrument (AYPA), which had sixty items. Hence, the more items in the tool, the less likely it is to be feasible and practical in these settings. Therefore the currently reviewed instruments, despite their importance in risk assessment, their lengthiest can be time-consuming in a limited resource setting with an overwhelming number of patients to attend [107].

A reliable instrument is needed to rapidly assess the clinical risks of exposure to ACEs for vulnerable children. The adoption of measures developed in HICs may not be feasible for LMICs. This is due to variances in ACEs and health outcomes based on socio-economic factors and diverse cultural and societal values. Furthermore, there are structural differences in the coordination of health systems between HICs and LMICs; therefore, there need to be different approaches to implementation. An evaluation study on Evidence-Informed Decision-Making (EIDM) in healthcare practice internationally illustrates multiple methodological pathways in adopting evidence interculturally. Therefore, its measurement can be challenged with policy bureaucratic of the desirable country, difficulties to establish counterfactuals, and long and undefined outcomes if practiced in other healthcare systems [108].

## Lack of instrument capacity to assess individual risk factors in biopsychosocial dimensions

Most of the reviewed instruments measured the vulnerability of children using a single cause-effect model to illustrate vulnerable children's ACEs, which is usually not expected in natural occurrence form. Consequently, it would have been ideal for the instruments to have monitored the impact of ACEs by addressing biopsychosocial dimensions and illustrating the multiple pathways of how the illness occurred and the overall management plan [98] For instance, an instrument such as African Youth Psychosocial Assessment Instrument (AYPA), which only assessed for psychosocial and behavioural challenges among African children affected by war, assessed internalising and externalising behaviours, such as prosocial behaviours (daily life functioning), depression, and anxiety attitudes. Child Psychosocial Distress Screener (CPDS) assessed for psychological aspects in four domains of distress, resilience, academic and intellectual capacity, school attendance, and IPAC, Instrument for Psychosocial Assessment for Childhood labor.

Usually, there is an interplay of one factor leading to the deterioration of another [109]. For example, a homeless child whose days unfolds on the street can acquire behavioural problems such as drug addiction and exposure to blood-borne diseases, including HIV/AIDS [110]. This includes the effects of poorly resourced neighborhoods that lack basic needs such as water, healthy food, and infrastructure [111]. The biopsychosocial model looks at health and illness due to both internal and external factors resulting from illness—internal factors related to biological manifestations such as physical characteristics related to genetics. External factors stem from behavioural factors associated with an individual's lifestyle, such as (daily stress level and health beliefs), social conditions (cultural influences, family relationships, social support). This will lead to deficiency in implementation due to the increased number of instruments for use

by frontline clinical professionals while decreasing the potential of integrating the biopsychosocial model of care in the current medical model of delivery care [103,112–114].

Although the current standard instrument for ACEs scores is still being debated whether there is a need to find effective treatments for effects and conditions resulting from ACEs [100,101,103]. In, LMICs, most of the poor health outcomes resulting from ACEs associated with poverty have already had current effective treatment. Therefore, screening for the risks associated with adversity in children is worthwhile for preventive measures and establishing referrals to adequate health and social support services [20,69]. For instance, impacts related to inadequate food capacity resulting in diseases related to under-nutrition can be effectively treated in the healthcare setting [20,32]. Behavioral problems resulting in drug addiction and promiscuity, which increase the risk of blood-borne and sexually transmitted diseases, can also be treated. Including social displacement and climatic disasters are likely to increase diseases that usually occur in dwelling areas such as refugee camps, including airborne diseases such as Tuberculosis, and diarrhea diseases, and typhoid fever due to poor sanitation in the refugee camps. Prompt detection of risks can enhance early management and treatment [115,116].

Despite that, the psychosocial measures are still under development for implementation. The efforts to strengthen partnerships with stakeholders will reduce exposure to ACEs and mediate the early impacts of childhood adversity. This can occur by facilitating better linkages within healthcare settings by collaborating through referral pathways that engage the multidisciplinary team. Furthermore, enhancing the continuous education within primary care settings to ensure that staff is trained and given the capacity to enhance referral mechanisms and intervene in partnership with behavioural health support services. In higher-income countries, progressive measures are conducted to integrate screening of adverse childhood experiences in a medical model of health care [103,117,118]. Through the CDC-Kaiser Permanente ACE Study, the Centres for Disease Control and Prevention (CDC) has been collating screening data that identifies the effects of adversity in children, referring them through established medical pathways to provide support and care for impacted individuals [24,42,112,119].

## Recommendations for implementation and policy

The primary health care systems in LIMC should consider ways to enhance the implementation and integration of the biopsychosocial aspect of care into the traditional biomedical model of health services. These measures will widen the consideration of holistic identification of adversity cause and effects in vulnerable and homeless children, compared to single effects measures as they are currently addressed. This includes working with community stakeholders of primary health care services such as NGOs (FBO) to forge a networking environment and referral mechanism for rehabilitation and social support services that are not available in the primary care settings. Furthermore, health care professionals need to be trained to address and advocate for the vulnerable population, including children who have complex needs. This can be achieved by alleviating stigma within primary care settings through training and proactive measures of integrating biopsychosocial care into traditional medical and treatment services. Therefore, healthcare professionals need to act as advocates, monitoring and establishing management pathways for adverse childhood experiences within healthcare settings. Following the process, indicators were recommended for rapid assessment of ACEs associated with poverty for vulnerable children for use by the frontline health care professionals in primary care settings.

## Limitations

We identified several limitations in the conduct of our literature review and narrative analysis. Firstly, there was a lack of a wide range of risk assessment instruments for monitoring adverse

and poverty-related risks for review in primary health care settings. Secondly, currently, no standardized instruments can be utilised as a gold standard for psychosocial issues, which decreased. There are no standardized instruments that can be utilized as a gold standard for psychosocial issues, which decreases the reliability and consistency of measurement instruments. Furthermore, there could be bias risk for selecting instruments for review, although we had a priori consensus established for the selected studies between the reviewers, which minimized the risk. Although, selected instruments for review might have a different interpretation based on diverse cultural perspectives and settings, which might influence the understanding of the instrument. Also, some instruments could not have been accessed due to the language barrier. Lastly, these findings may be overemphasized due to the nature of narrative systematic review analysis; therefore, it may not be feasible for replication in other settings.

## Conclusion

In this systematic review, we aimed to find an instrument or individual indicators or items of instruments suitable for the rapid assessment of biopsychosocial effects of poverty-related adversity in vulnerable and homeless children for frontline clinical health care professionals in Lower Middle-Income Countries.

The findings indicate that the identified instruments lack tracking measures for adverse childhood experiences in frontline clinical settings as they were created for use in non-front line clinical settings. Healthcare systems in these countries have delayed integrating the psychosocial measures into a biomedical treatment model; hence, healthcare professionals are not yet ready to track social dimensions in their routine health services. There is a need for healthcare systems to commit to infrastructure and resources to equip monitoring systems for early warning signs of adverse childhood experiences through a functional multidisciplinary system and to enhance referrals and networking to maximize equity for vulnerable populations.

## Supporting information

**S1 Checklist. PRISMA checklist.**
(DOCX)

**S1 Table. Characteristics of instruments to assess adversities related to poverty in Low- and Middle-Income Countries (LMICs).**
(DOCX)

**S2 Table. Risk assessment tool ranking for use in practice using a 10-point checklist for external validation measures.**
(DOCX)

**S3 Table. Risk assessment tool ranking for use in practice using a 10-point checklist.**
(DOCX)

**S4 Table. The instrument's feasibility, biopsychosocial dimension, and applicability for use in frontline clinical settings.**
(DOCX)

**S5 Table. A Measurement instrument to assess biopsychosocial indicators related to poverty.**
(DOCX)

**S6 Table. (Proposed for) rapid assessment of the biopsychosocial dimension of childhood adversity associated with poverty.**
(DOCX)

## Acknowledgments

The author wishes to thank Josephine McGill. Shannon Brown and Leila Mohammad from the Flinders University, Library, for their help during the literature search process.

## Author Contributions

**Conceptualization:** Winfrida Mwashala.

**Data curation:** Winfrida Mwashala, Udoy Saikia, Diane Chamberlain.

**Formal analysis:** Winfrida Mwashala, Diane Chamberlain.

**Investigation:** Winfrida Mwashala, Udoy Saikia, Diane Chamberlain.

**Methodology:** Winfrida Mwashala, Udoy Saikia, Diane Chamberlain.

**Project administration:** Winfrida Mwashala.

**Supervision:** Udoy Saikia, Diane Chamberlain.

**Writing – original draft:** Winfrida Mwashala.

**Writing – review & editing:** Winfrida Mwashala, Udoy Saikia, Diane Chamberlain.

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
