## [Decision Letter · Decision Letter 0]

2 Nov 2021

PGPH-D-21-00734

Screening instruments to assess the biopsychosocial effects of Adverse Childhood Experiences for vulnerable and homeless children in low- and middle-income countries: A systematic review and narrative synthesis

Dear Dr. Mwashala,

Thank you for submitting your manuscript to PLOS Global Public Health. After careful consideration, we feel that it has merit but does not fully meet PLOS Global Public Health’s publication criteria as it currently stands. Therefore, we invite you to submit a revised version of the manuscript that addresses the points raised during the review process.

We look forward to receiving your revised manuscript.

Kind regards,

Rohina Joshi

Academic Editor

Journal Requirements:

1. Please update the completed 'Competing Interests' statement, including any COIs declared by your co-authors. If you have no competing interests to declare, please state "The authors have declared that no competing interests exist". Otherwise please declare all competing interests beginning with the statement "I have read the journal's policy and the authors of this manuscript have the following competing interests:"

2. Tables should not be uploaded as individual files.  Please remove these files and include the tables in your manuscript file.

Reviewers' comments:

Reviewer's Responses to Questions

**Comments to the Author**

1. Does this manuscript meet PLOS Global Public Health’s publication criteria? Is the manuscript technically sound, and do the data support the conclusions? The manuscript must describe methodologically and ethically rigorous research with conclusions that are appropriately drawn based on the data presented.

Reviewer #1: Partly

Reviewer #2: Partly

2. Has the statistical analysis been performed appropriately and rigorously?

Reviewer #1: Yes

Reviewer #2: Yes

3. Have the authors made all data underlying the findings in their manuscript fully available (please refer to the Data Availability Statement at the start of the manuscript PDF file)?

Reviewer #1: Yes

Reviewer #2: Yes

4. Is the manuscript presented in an intelligible fashion and written in standard English?

Reviewer #1: Yes

Reviewer #2: Yes

5. Review Comments to the Author

Reviewer #1: General

Any exploration of adverse childhood experiences (ACEs) in low/middle income countries is welcome, but it is a bit strange that this review- is looking only at screening tools that assess the biopsychosocial effects of ACEs in an already high risk population- i.e., vulnerable and homeless. Why? Surely there is quite a literature specifically on homeless and street children, similarly a lot of indicators (morbidity/mortality) that vulnerable child populations- ie those that are living in socially adverse circumstances have higher rates of morbidity and mortality. It is not clear at the outset what the authors mean by “vulnerable”- not defined clearly- seems to pertain to children living in poverty. Similarly, the term “biopsychosocial” effects is not defined. I am afraid that I could only download the pdf version of the submission through Microsoft Edge-none of the tables actually were visible- so I may have missed the clarity provided by those. Nevertheless the premise of this systematic review is flawed at the outset. I would suggest that a rewrite with title: Review of the tools used to identify and respond to adverse childhood experiences in vulnerable child and youth populations in LMICs- a much better and useful contribution. Or if the authors intention was to review assessment tools that documented bio-psychosocial effects of exposure to social adversity, then that is the title, ie: Screening instruments to assess the biopsychosocial effects of exposure to social adversity in vulnerable child/youth populations in LMIC. There are quite a few edits needed in language and grammar, such as frequent misuse of singular where plurals should be used (such as instrument for instruments)- but that can be responded to later.

Abstract

This should be re-written following the major re-write required. ACEs and ACES are used interchangeably- please use ACEs or just ACE.

Introduction

The major concern is that it is not clear who the target population is. If it is children living in poverty and extreme social disadvantage- as the authors seem to imply-then it needs to be specified and justified. There is some vague reference to poverty-associated adversity and then also to HIV and displaced populations, but no clear focus on children in LMIC who face extreme disadvantage such as street children, those who are deemed orphan and vulnerable children. This really screams out to be clarified!

Also the authors claim that ACE screening in HICs mainly uses the medical model- this is a complete misrepresentation. Mostly ACEs research has focused on adult populations and only recently has it come to be used actively as a tool for screening and assessment in child/youth populations. There is plenty of robust concern about screening for ACEs already articulated in HICs.

These are the aims

i) What are the existing screening and monitoring systems and instruments for identifying biopsychosocial effects of ACES in frontline clinical settings in LMICs?

ii) ii) For the identified screening instruments, what are their feasibility and applicability for assessing vulnerable children that are exposed to ACES in a rapid manner? 7

iii) iii) What patterns of adverse childhood indicators need to be screened for vulnerable and homeless children in healthcare settings to include the biopsychosocial dimensions of ACEs?

Given that the authors have not mentioned or defined bio-psychosocial effects, it seems a bit presumptuous to be looking at tools that record this. Also who are vulnerable/homeless children- surely it would be much more useful to look at tools that can be used for all children who present to frontline health services in LMICs so that ACEs can be identified and responded to? A re-write of the aims to review tools and instruments to identify ACEs and address them in high risk, vulnerable children/youth (however defined) in LMIC- or review tools/instruments to explore the biopsychosocial effects of adversity would be better.

Methods

I am happy that the authors have used robust methodology for their review and I can’t fault that. The problem lies in the aims of the review.

Results

“In stage two of the review of the identified instrument” what does this mean?? I think this is about measurement tools and instruments. Really unclear.

Will require a re-write, once the aims and review terms are clarified.

Discussion

Will require a re-write, once the aims and review terms are clarified.

Reviewer #2: I have attached my comments in the attached and uploaded word document. I have also copied it here

Screening instruments to assess the biopsychosocial effects of Adverse Childhood

Experiences for vulnerable and homeless children in low- and middle-income

countries: A systematic review and narrative synthesis

The authors attempt to address this important issue of appropriate early identification of ACE in children in LMIC. They have done this by identifying the currently published instruments in selected countries and comparing them using mixed methods and narrative methodologies, concluding that no single instrument in use comprehensively does what it is meant to do. In addition it appears that the authors also express their views on what causes adversity in children in LMIC, how this is different from children in HIC and what kind of training the primary health workers in LMIC need in order to achieve this objective.

While the aims and objectives and methodologies employed are appropriate, there are several areas which can be improved in its execution.

• Overall the paper is a slightly laborious read, with the themes repeating multiple times in different sections of the manuscript with the second part of their aims only partially achieved.

• There are a number of grammatical errors which need to be fixed including uses of tense – for example when the authors talk of multiple scales, they refer to it in singular which confuses the reader.

• The objective is also unclear at times – are they trying to highlight the items that need to go into the development of a new instrument to measure ACE in LMIC or are they merely comparing current instruments? It is better to separate these objectives clearly with the latter mentioned or discussed in the future directions section as it is not a stated aim.

• In the methods section, they mention the prospero number is awaited – needs to be updated

• Table one – why the search term included only Africa as the geographical location is unclear

• Table two – the criteria is good, however it does not seem that relevant literature from all MLIC were included

• Overall the tables have good information about the studies and the instruments used, however there is almost no mention about the individual studies or instruments in the body of the paper. It would be preferable if some cross referencing was made, esp in the discussion section

• The section on external validation could be tightened

• While the validation sections give good information, they remain as siloed pieces of information and their relevance is lost

• Para two of discussion has poor sentence construction. And throughout the section, ‘instrument’ is used instead of ‘instruments’ and presumably the authors are speaking about all of them in the identified studies.

• Towards the end of para three of discussion, they make reference to identifying indicators for early warning signs – not clear what they mean by this phrase

• Para four of discussion is confusing as it is not clear if they talk about their study or something else and its relevance to their study

• The last sentence of the recommendations section needs to be reconstructed as it is not clear what the authors are trying to express.

• A number of important limitations has been missed

o Not enough countries are chosen to represent all the diversity in LMIC – for example there is no reference to India and the Middle east and also South East Asia and South America which together has over half the population of LMIC with incredible diversity

o No acknowledgement of the various cultural and linguistic nuances and adaptations that will be necessary for these instruments to be effective and utilized to achieve their logical objective

o Also unclear if the instruments are only measuring symptoms or whether they throw some light into the ‘cause’ and ‘effect’ of ACEs

o Also unclear if these instruments measure clinical risk which can help in the further disposition and response

• Referencing and discussing a number of other relevant studies in the field would further enrich this study. Some of these include a smattering of studies from other LMIC. It is not clear if these studies were included in the initial search

o Fernandes et al. BMJ open 2021 – talk about a protocol to access child maltreatment in India

o Seshadri S et al. NIMHANS child project (India) has a wealth of information in its repository which is very relevant to this paper with adaptations of various scales in local languages and dialects in India

o Escueta et al. in BMC int human rights (2014) discussed effects of ACEs on disadvantaged children from five LEC

o Al Shawi et al. in BMC public health (2019) reported effects of ACEs on young people in Iraq and Almuneef et al did the same in Child Abuse and Neglect (2014) from Saudi Arabia

o Tran et al (2015) reported on the impact of ACEs on students from Vietnam and Ramaiya from Indonesia reported on similar issues in young adolescents in J Adolesc Health (2021)

6. PLOS authors have the option to publish the peer review history of their article (what does this mean?). If published, this will include your full peer review and any attached files.

**Do you want your identity to be public for this peer review?** For information about this choice, including consent withdrawal, please see our Privacy Policy.

Reviewer #1: No

Reviewer #2: No

---

## [Decision Letter · Decision Letter 1]

2 Aug 2022

Instruments to identify risk factors associated with Adverse Childhood Experiences for vulnerable children in primary care in Low- and Middle-Income Countries: A systematic review and narrative synthesis

PGPH-D-21-00734R1

Dear Mwashala,

We are pleased to inform you that your manuscript 'Instruments to identify risk factors associated with Adverse Childhood Experiences for vulnerable children in primary care in Low- and Middle-Income Countries: A systematic review and narrative synthesis' has been provisionally accepted for publication in PLOS Global Public Health.

Best regards,

Julia Robinson

Staff Editor

Reviewer Comments (if any, and for reference):

Reviewer's Responses to Questions

**Comments to the Author**

1. If the authors have adequately addressed your comments raised in a previous round of review and you feel that this manuscript is now acceptable for publication, you may indicate that here to bypass the “Comments to the Author” section, enter your conflict of interest statement in the “Confidential to Editor” section, and submit your "Accept" recommendation.

Reviewer #2: All comments have been addressed

2. Does this manuscript meet PLOS Global Public Health’s publication criteria? Is the manuscript technically sound, and do the data support the conclusions? The manuscript must describe methodologically and ethically rigorous research with conclusions that are appropriately drawn based on the data presented.

Reviewer #2: Yes

3. Has the statistical analysis been performed appropriately and rigorously?

Reviewer #2: Yes

4. Have the authors made all data underlying the findings in their manuscript fully available (please refer to the Data Availability Statement at the start of the manuscript PDF file)?

Reviewer #2: Yes

5. Is the manuscript presented in an intelligible fashion and written in standard English?

Reviewer #2: Yes

6. Review Comments to the Author

Reviewer #2: It is pleasing to note that the authors have accepted most of the comments and as a result, the paper is succinct and clear.

The only part which I was still unsure of was the criteria of selection of studies from only a few LMIC and leaving out many others.

7. PLOS authors have the option to publish the peer review history of their article (what does this mean?). If published, this will include your full peer review and any attached files.

**Do you want your identity to be public for this peer review?** For information about this choice, including consent withdrawal, please see our Privacy Policy.

Reviewer #2: **Yes: **A/Prof Rajeev Jairam
